

# Is there a link between aging and microbiome diversity in exceptional mammalian longevity?

Graham M. Hughes[1], John Leech[2], Sébastien J. Puechmaille[1,3], Jose V. Lopez[4] and Emma C. Teeling[1]

[1] School of Biology and Environmental Science, University College Dublin, Dublin, Ireland
[2] APC Microbiome Institute, University College Cork, Cork, Ireland
[3] Zoology Institute, Greifswald, Germany
[4] Halmos College of Natural Sciences and Oceanography, Nova Southeastern University, Dania Beach, FL, United States of America

## ABSTRACT

A changing microbiome has been linked to biological aging in mice and humans, suggesting a possible role of gut flora in pathogenic aging phenotypes. Many bat species have exceptional longevity given their body size and some can live up to ten times longer than expected with little signs of aging. This study explores the anal microbiome of the exceptionally long-lived *Myotis myotis* bat, investigating bacterial composition in both adult and juvenile bats to determine if the microbiome changes with age in a wild, long-lived non-model organism, using non-lethal sampling. The anal microbiome was sequenced using metabarcoding in more than 50 individuals, finding no significant difference between the composition of juvenile and adult bats, suggesting that age-related microbial shifts previously observed in other mammals may not be present in *Myotis myotis*. Functional gene categories, inferred from metabarcoding data, expressed in the *M. myotis* microbiome were categorized identifying pathways involved in metabolism, DNA repair and oxidative phosphorylation. We highlight an abundance of 'Proteobacteria' relative to other mammals, with similar patterns compared to other bat microbiomes. Our results suggest that *M. myotis* may have a relatively stable, unchanging microbiome playing a role in their extended 'health spans' with the advancement of age, and suggest a potential link between microbiome and sustained, powered flight.

## INTRODUCTION

The importance of the gut microbiome, the collection of microflora or microbiota inhabiting various regions of the gastro-intestinal tract, has become apparent in recent years. It is known to facilitate the fermentation of nutrients, such as carbohydrates, into short chain fatty acids (e.g., butyrate) for use by the host (*Flint et al., 2012*). Many vitamins are also synthesized by the microbiome such as vitamin B7, B12 and Vitamin A, which may be otherwise unavailable (*O'Hara & Shanahan, 2006*). In addition to nutrition,

Corresponding author
Emma C. Teeling,
emma.teeling@ucd.ie

more evidence is accumulating pointing to "microbiome-wide associations" with health and disease in humans and other hosts (*Gilbert et al., 2016*), and the composition of the mammalian gut microbiome has been linked to the process of biological aging (*Biagi et al., 2016*; *Mello et al., 2016*). Aging is characterized by the progressive decline of function, increased frailty and an increase in chronic disease (*López-Otín et al., 2013*). Studies of the human microbiome have reported shifts in microbial composition across different stages of life, with a high degree of variability at the two extremes of infancy and old age (*Saraswati & Sitaraman, 2015*). A shift from a microbiome that deals exclusively with breast milk in nursing infants to a more diverse microbiome that can metabolize a wider range of nutrition (*Yatsunenko et al., 2012*) is observed in early to middle stages of life. In later life, a reduction of lactobacilli and an increase of potentially pathogenic *Enterobacteriaceae* have been observed in frail individuals (*Van Tongeren et al., 2005*). This accumulation of pathogenic flora has been associated with a range of clinical problems such as infection, cancer and deficiencies in immune response (*Atarashi et al., 2013*; *Saraswati & Sitaraman, 2015*). Similar microbial shifts are observed in mice, such as the decrease in bacteria that synthesize vitamin B12 in older age cohorts, leading to overall changes in microbiome composition and function in age related frailty (*Langille et al., 2014*) implying a general trend in the aging gut. The question of whether or not such microbial shifts are a symptom rather than a driver of aging has yet to be conclusively answered.

Bats are exceptional mammals not only because of their capability of powered flight but also due to the diverse range of life histories they exhibit, with exceptional longevity being of particular interest (*Austad, 2010*). Within bats, a number of species in the family Vespertilionidae demonstrate extreme longevity, living up to ten times longer than expected given their body size (*Austad, 2010*; *Shen et al., 2010*). Surviving in the wild requires maintaining agility; speed and high frequency hearing to capture prey on a daily basis. Therefore, a long lifespan in bats coincides with a long health-span. Elucidating the changes that occur in microbial composition over time in these exceptionally long-lived organisms will shed light on the role of the microbiome in extended health-spans. While previous studies of bat microbiomes have focused on the effects of phylogeny and diet, this is the first bat microbiome study to focus on aging.

In this study, we have used DNA metabarcoding of the 16S rRNA gene and high-throughput sequencing to characterize the structure and function of the anal microbiome from 52 wild, exceptionally long–lived insectivorous *Myotis myotis* bats (maximum lifespan (MLS) 37 years *Gaisler et al., 2003*) using a non-lethal sampling method. *M. myotis* makes an ideal comparison to current mouse microbiome models as these bats are similar in body size but can live up to 32 years longer. We show that there is no obvious shift in bacterial composition between juvenile and the early stages of adulthood in *M. myotis*, as has been observed in other mammals. We find that the highest number of pathways expressed in bat metagenomes are involved in metabolism, energy consumption, DNA repair and oxidative phosphorylation, all reactions that may play a role in the use of powered flight and aging, suggesting a potential interaction between long-term microbiome stability, lifespan and powered flight. A microbiome that does not change over time may have profound effects on age-related infections and immune deficiencies, opening up new avenues of microbiome

research in non-model organisms. Finally, we compare the relative abundance of major bacterial phyla in *M. myotis* with those of other bats and mammals, highlighting the high abundance of 'Proteobacteria' in Chiroptera relative to other mammals.

## MATERIALS AND METHODS

### Sample collection

All field procedures were carried out in accordance with the ethical guidelines and permits delivered by 'Arrêté' by the Préfet du Morbihan, Bretagne awarded to Eric Petit, Frédéric Touzalin and Sébastien Puechmaille for the time period 15 June-15 September 2013–2017. Full ethics approval and permission (AREC-13-38-Teeling) for capture and field sampling was awarded by the University College Dublin ethics committee. *M. myotis* were sampled in western France, (Brittany), July 2013, from four large roost locations: Béganne (Beg), Férel (Fer), La Roche-Bernard (LRB) and Noyal-Muzillac (NM). The maximum distance between any of the four sites is roughly 24 km. Bats were caught in custom harp traps while leaving their roost (hence typically before foraging/feeding) and were initially placed in individual cloth bags (*Huang et al., 2016*). Each bat was identified by a unique transponder inserted under the skin when the bat was first captured. If captured and transponded as a juvenile (∼<6 weeks old), indicated by the lack of fused finger bones, the exact age at recapture was known. Validated weaning status could not be determined for juvenile bats, however the sampled juveniles were capable of flight, suggesting that the majority were partially weaned, but opportunistic suckling cannot be ruled out. If caught and transponded for the first time as an adult, years since first capture was noted and a plus sign indicated the individual was older than age estimated from first capture. For each bat, a Copan FLOQSwab™ swab was gently inserted into the anus and removed. The swab was then placed into an Eppendorf and immediately flash frozen in liquid nitrogen. Swabs were subsequently place in ethanol during transport for sequencing.

### Categorical datasets

Sample age cohorts ranged from '0 years' to ' 4+ years', with individuals whose exact age was unknown but minimum age could be determined denoted by '+'. To investigate differences in anal flora between juvenile and adult bats, samples were categorized into a number of age data sets. Age dataset 1 contained known ages only ($n = 29$). When comparing adults and definitive juveniles, age dataset 2 categorized bats that were 1 or more years old as adults ($n = 33$), while 0 year olds were considered juvenile ($n = 19$). In age dataset 3, the cut–off for juvenile was increased from 0 to 1 year (juvenile $n = 27$), to investigate if the microbiome of a 1 year old *M. myotis* bat transitioning from juvenile to adult had an influence on any age associated microbial shifts that might be observed. For age dataset 3, ' 1+' individuals were excluded. Finally, age dataset 4 contained all ages as individual categories (Table 1). Juvenile to adult age ranges cover the entire lifespan of a mouse or similar sized mammal (0–4 years). In addition to age, bat samples were categorized based on gender and location of collection (sample site was not catalogued for two individuals, MMY247 and MMY51, which were instead referred to as 'site undefined').

**Table 1  Sample information.** Sample (MMY + sample number) metadata describing different numerical and categorical variables are displayed. The number of reads sequenced for each sample, and subsequent alpha diversity measures (PD Whole tree and observed OTUs) are counted. Age datasets were used to compare individual ages and to compare juveniles and adults.

| #Sample ID | Sex | Age dataset 1 | Age dataset 2 | Age dataset 3 | Age dataset 4 | Site | # Reads | PD whole tree | Observed OTUs |
|---|---|---|---|---|---|---|---|---|---|
| MMY1028 | F | 0 | Juvenile | Juvenile | 0 | LRB | 75,913 | 7.81 | 77 |
| MMY144 | F | NA | Adult | Adult | 4+ | Beg | 30,005 | 25.03 | 242 |
| MMY18 | M | NA | Adult | NA | 1+ | LRB | 136,568 | 28.16 | 421 |
| MMY19 | M | NA | Adult | NA | 1+ | LRB | 243 | 26.59 | 352 |
| MMY21 | F | NA | Adult | Adult | 2+ | LRB | 684 | 29.75 | 446 |
| MMY230 | F | 3 | Adult | Adult | 3 | Fer | 43,623 | 15.28 | 184 |
| MMY231 | M | NA | Adult | Adult | 3+ | Fer | 1911 | 17.65 | 183 |
| MMY233 | F | NA | Adult | Adult | 3+ | Fer | 52,169 | 18.68 | 289 |
| MMY247 | F | NA | Adult | Adult | 4+ | Site undefined | 58,855 | 25.07 | 322 |
| MMY272 | F | 1 | Adult | Juvenile | 1 | Fer | 275 | 31.98 | 470 |
| MMY329 | F | 1 | Adult | Juvenile | 1 | Fer | 140,485 | 33.53 | 411 |
| MMY332 | M | 0 | Juvenile | Juvenile | 0 | Fer | 97 | 17.14 | 182 |
| MMY387 | M | NA | Adult | Adult | 2+ | NM | 107,414 | 36.85 | 560 |
| MMY388 | F | NA | Adult | NA | 1+ | NM | 104,612 | 19.21 | 280 |
| MMY391 | M | NA | Adult | Adult | 2+ | NM | 42,732 | 10.92 | 114 |
| MMY51 | F | NA | Adult | NA | 1+ | Site undefined | 66,819 | 7.61 | 103 |
| MMY519 | F | NA | Adult | NA | 1+ | NM | 49,322 | 30.86 | 489 |
| MMY52 | F | NA | Adult | NA | 1+ | LRB | 10,971 | 15.55 | 141 |
| MMY524 | M | 1 | Adult | Juvenile | 1 | NM | 52,755 | 22.97 | 313 |
| MMY540 | F | NA | Adult | NA | 1+ | NM | 586 | 32.86 | 441 |
| MMY573 | F | NA | Adult | Adult | 4+ | Beg | 134 | 27.65 | 350 |
| MMY583 | F | 0 | Juvenile | Juvenile | 0 | Beg | 76,738 | 23.03 | 368 |
| MMY589 | F | 0 | Juvenile | Juvenile | 0 | Beg | 128,039 | 21.09 | 179 |
| MMY729 | M | 1 | Adult | Juvenile | 1 | Fer | 85,598 | 15.35 | 213 |
| MMY730 | F | 1 | Adult | Juvenile | 1 | Fer | 165,854 | 14.01 | 108 |
| MMY731 | F | NA | Adult | Adult | 2+ | Fer | 305 | 23.40 | 315 |
| MMY732 | F | NA | Adult | Adult | 3+ | Fer | 1,173 | 10.60 | 136 |
| MMY736 | M | 3 | Adult | Adult | 3 | Fer | 52 | 30.48 | 428 |
| MMY750 | M | NA | Adult | NA | 1+ | Fer | 46,504 | 7.55 | 163 |
| MMY780 | F | 0 | Juvenile | Juvenile | 0 | NM | 34,536 | 27.65 | 399 |
| MMY781 | M | 0 | Juvenile | Juvenile | 0 | NM | 44,487 | 18.57 | 165 |
| MMY808 | M | NA | Adult | Adult | 2+ | NM | 79,976 | 38.83 | 554 |
| MMY810 | M | NA | Adult | Adult | 2+ | NM | 498 | 17.23 | 202 |
| MMY835 | M | 1 | Adult | Juvenile | 1 | LRB | 91,311 | 9.32 | 84 |
| MMY840 | M | NA | Adult | Adult | 2+ | LRB | 55,681 | 10.791 | 137 |
| MMY841 | F | 0 | Juvenile | Juvenile | 0 | LRB | 44,925 | 20.93 | 267 |
| MMY860 | F | 1 | Adult | Juvenile | 1 | NM | 39,929 | 26.58 | 361 |

| #Sample ID | Sex | Age dataset 1 | Age dataset 2 | Age dataset 3 | Age dataset 4 | Site | # Reads | PD whole tree | Observed OTUs |
|---|---|---|---|---|---|---|---|---|---|
| MMY863 | M | 0 | Juvenile | Juvenile | 0 | NM | 106,015 | 21.11 | 312 |
| MMY864 | F | NA | Adult | Adult | 2+ | NM | 9,608 | 25.91 | 302 |
| MMY865 | M | 0 | Juvenile | Juvenile | 0 | NM | 62,705 | 25.22 | 353 |
| MMY883 | F | 0 | Juvenile | Juvenile | 0 | NM | 59,037 | 47.05 | 747 |
| MMY887 | F | 0 | Juvenile | Juvenile | 0 | LRB | 99,393 | 7.81 | 77 |
| MMY893 | F | NA | Adult | NA | 1+ | NM | 56,760 | 25.03 | 242 |
| MMY894 | M | 0 | Juvenile | Juvenile | 0 | NM | 82,872 | 28.16 | 421 |
| MMY930 | M | 0 | Juvenile | Juvenile | 0 | LRB | 83,416 | 26.59 | 352 |
| MMY932 | M | 0 | Juvenile | Juvenile | 0 | LRB | 191,120 | 29.75 | 446 |
| MMY937 | M | 1 | Adult | Juvenile | 1 | NM | 12,983 | 15.28 | 184 |
| MMY942 | M | 0 | Juvenile | Juvenile | 0 | NM | 60,449 | 17.65 | 183 |
| MMY943 | F | 0 | Juvenile | Juvenile | 0 | NM | 15,264 | 18.68 | 289 |
| MMY946 | F | 0 | Juvenile | Juvenile | 0 | NM | 118,085 | 25.07 | 322 |
| MMY987 | M | 0 | Juvenile | Juvenile | 0 | NM | 25,295 | 31.98 | 470 |
| MMY990 | F | 0 | Juvenile | Juvenile | 0 | NM | 93,196 | 33.53 | 411 |

## 16S rRNA library preparation and sequencing

Total microbial genomic DNA from each swab sample was extracted using MO BIO's PowerLyzer™ PowerSoil® kit, following Earth Microbiome (http://earthmicrobiome.org) standards and recommendations. Following extractions, DNA quality was checked via agarose gel electrophoresis. Purity of the 16S rRNA amplicons after adding barcodes was determined via a 2200 TapeStation Bioanalyzer (Agilent Technologies, Santa Clara, CA, USA), while quantity of each DNA (4 pM of each sample) was measured with a Qubit 2.0 flourometer (Life Technologies, Carlsbad, CA, USA) prior to each MiSeq run. The protocol detailed in *Caporaso et al. (2012)*, using the V4 hyper-variable region primers 515F/806R for paired-end 16S rRNA sequencing was strictly followed.

Libraries were generated for all 52 samples using the Illumina Nextera XT dual primer protocol and 16S metagenomic library prep guide. Blank extractions containing no DNA, and tested with barcoded and non-barcoded index primers, were used as a negative control to investigate potential contamination during laboratory procedures. Samples were sequenced using the MiSeq platform and 250bp paired end chemistry. Taxonomic analysis was carried out using the Quantitative Insights into Microbial Ecology (QIIME) suite of software, version 1.9.1 (*Caporaso et al., 2010*). Paired end reads were filtered using a phred score threshold of 30, and chimeric sequences were removed using the QIIME '*split_libraries_fastq.py*' and '*filter_fasta.py*' scripts. Reads from each sample were then overlapped, joining the forward and reverse reads, using the fastq-join method as implemented in QIIME.

## OTU Picking and diversity analyses

Overlapped sequences from each bat sample were used for Operational Taxonomic Unit (OTU) picking. OTU picking involves the process of aligning sequences found in each sample to a reference database and assigning them to an OTU (a cluster of similar reads

representing a bacterial phyla, genera or species, depending on level of resolution). For this study, the Greengenes annotated 16S reference database (*DeSantis et al., 2006*) and PyNast aligner was used, with a minimum sequence clustering identity of 97% (*Caporaso et al., 2010*). Greengenes was chosen as our reference database as it has been previously used to characterize other bat microbiomes, which are included in our study (*Phillips et al., 2012*; *Carrillo-Araujo et al., 2015*). The method of 'open OTU picking', where reads that do not map to a reference database are subsequently clustered *de novo*, was applied.

The mean OTU abundance within each sample was calculated as alpha diversity in QIIME. Samples were rarefied to 8,000 reads, removing possible biases introduced due to uneven sequencing depth, using the '*core_diversity_analysis.py*' script, with parameter '–e' set to 8,000. This minimum was chosen to include as many samples as possible while also providing enough depth to get a good representation of the microbiome. Despite potentially reduced statistical power (*McMurdie & Holmes, 2014*), rarefaction was chosen as the ideal means of data normalization given the range of reads across samples (*Weiss et al., 2017*), resulting in abundances comparable to other mammalian microbiomes (see below). A change in alpha diversity has previously been used to differentiate between young, mature and old mammals (*Yatsunenko et al., 2012*; *Frese et al., 2015*), hence alpha-diversity was calculated across each pre-defined age dataset, in addition to gender and location of collection. The mean phylogenetic diversity (PD), a measure of alpha diversity accounting for phylogenetic differences between bacterial species (*Faith, 1992*), was calculated for each sample and metadata category. This was compared to the gut flora of previously characterized members of Vespertilionidae (*Phillips et al., 2012*) to determine if the anal flora is representative of the gut microbiome, further implying the utility of our sampling method. Beta diversity (abundance between samples) was also calculated across all samples before and after rarefaction using both weighted and un-weighted Unifrac distance matrices, which are based on differences between samples using phylogenetic information (*Lozupone & Knight, 2005*). Hierarchical clustering, using Ward's method (*Ward Jr, 1963*), based on number of OTUs detected was applied to samples using *R* to visualize any separation based on age, sex or location that could be observed. The core microbiome was computed as bacterial species present in at least a 'user-specified' percentage of samples. Minimum thresholds ranging from 50–80% were investigated across each age dataset, gender and location.

## Statistical analysis

PCoA plots were generated based on the weighted Unifrac distance matrices of beta-diversity for each different categorical variable. These plots were visualized to determine if linear separation existed between ages, gender and location of sample collection. Additionally, Analysis of Similarity (ANOSIM) was applied to the weighted Unifrac distance matrices to determine if a statistically significant difference existed for each data category. ANOSIM is non-parametric and was made distribution-free by using a number of permutations (9,999).

OTU frequencies between juvenile and adult samples were compared using a Kruskal–Wallis analysis of variance to identify bacteria whose abundance was significantly different

across categories, using the '*group_significance.py*' script in QIIME, with Bonferroni correction applied across all results. Additionally, Similarity Percentage (SIMPER; *Clarke, 1993*) analysis, a method to assess taxa responsible for the overall observed similarity/dissimilarity between groups, was carried out on the pooled OTU abundance counts for each location both before and after rarefaction. SIMPER was carried out using the PAST software (*Hammer, Harper & Ryan, 2001*).

## Predictive characterization of microbiomes (PICRUSt)

Phylogenetic Investigation of Communities by Reconstruction of Unobserved States (PICRUSt; *Langille et al., 2013*) was used to predict genes/ pathways that might be expressed in bacteria found in the anus of *M. myotis*. Based on bacterial species whose protein coding genes are described in the KEGG and COG databases, PICRUSt attempts to infer genes expressed in each OTU using phylogenetic similarity to a previously characterized species. As per its operational requirements, PICRUSt was used on a normalized biom table generated using the 'closed OTU picking' method in QIIME, excluding reads that could not be mapped to a reference OTU cluster. To investigate if metagenomic changes occurred between juvenile and adult microbiomes, functional analysis was conducted using the methodologies described by *Phillips et al. (2017)*, implemented in the R package '*FunkyTax*'. This analysis was used to identify if predicted metagenomic function remained the same, were enhanced (increase in abundances of contribution across age groups) or divergent (significantly different composition across ages for a predicted function).

## Phylogenetic comparison to other mammals

The average relative abundances of nine bacteria phyla in *M. myotis* were compared to available microbiome data from other mammals, focusing specifically on the gut, rectal or faecal samples, to investigate if closely related taxa shared similar composition. Bacterial abundances were compared under the assumption that, while based on different experimental conditions and research goals, each mammalian microbiome study represents an accurate abundance calculation for that species given the experimental setup. The relative abundances of 35 additional bat species (*Phillips et al., 2012*; *Carrillo-Araujo et al., 2015*), cow rectum (*Mao et al., 2015*), dolphin rectum (*Bik et al., 2016*), dog faecal sample (*Swanson et al., 2011*), human and mouse faecal sample (*Krych et al., 2013*), gorilla faecal sample (*Gomez et al., 2015*) and Tasmanian devil faecal samples (*Cheng et al., 2015*) were compared using principal component analysis (PCA). $K$-means clustering, with the optimum $k$ number of clusters decided using the elbow criterion, was carried out in $R$ and used to identify if any 'bat-specific' clusters were present, and if species clustering resembled phylogeny (*Datzmann, Helversen & Mayer, 2010*; *Meredith et al., 2011*; *Phillips et al., 2012*).

## RESULTS

### QIIME analyses

Across all 52 samples, 2,947,977 read pairs were overlapped and used for downstream analyses (Table 1). No spurious amplifications were detected in the negative controls. A

**Table 2  Phylogenetic diversity within the *Myotis myotis* microbiome.** Mean Phylogenetic Diversity (PD) for each category, representing the range of bacterial species present, are shown for each data category.

| Age dataset 1 | 0 years | 1 year | 3 years | Unknown | | | |
| --- | --- | --- | --- | --- | --- | --- | --- |
| | 19.36 | 21.24 | 26.58 | 26.14 | | | |
| Age dataset 2 | Adults | Juveniles | | | | | |
| | 24.67 | 19.36 | | | | | |
| Age dataset 3 | Adults | Juveniles | Unknown | | | | |
| | 24.01 | 19.88 | 28.94 | | | | |
| Age dataset 4 | 0 Years | 1 Year | 1+ Years | 2+ Years | 3 Years | 3+ Years | 4+ Years |
| | 19.36 | 21.24 | 28.94 | 23.90 | 26.59 | 29.75 | 20.15 |
| Gender | Male | Female | | | | | |
| | 20.4 | 23.85 | | | | | |
| Site | Beg | Fer | LRB | NM | | | |
| | 14.52 | 23.89 | 18.00 | 24.67 | | | |

**Table 3  Core microbiomes.** Bacteria phyla and the number of OTUs present in a range of core microbiome percentage thresholds for *M. myotis* are displayed.

| | Core 50% | Core 60% | Core 70% | Core 80% | Core 90% |
| --- | --- | --- | --- | --- | --- |
| Number of OTUs | 47 | 29 | 20 | 13 | 7 |
| % Proteobacteria | 57.45% | 55.17% | 50% | 38.46% | 42.86% |
| % Actinobacteria | 17.02% | 17.24% | 20% | 15.38% | 14.28% |
| % Firmicutes | 17.02% | 20.69% | 30% | 46.16% | 42.86% |
| % Cyanobacteria | 4.25% | 3.45% | 0% | 0% | 0% |
| % Chlamydiae | 2.13% | 0% | 0% | 0% | 0% |
| % Tenericutes | 2.13% | 3.45% | 0% | 0% | 0% |

rarefying threshold of 8,000 sequences reduced the final number of *M. myotis* samples to 41 individuals. The number of unique OTUs per sample ranged from 77 to 747 (mean = 299). No age, sex or location-specific clusters were observed across samples using hierarchical clustering (Fig. S1). Mean alpha diversity (diversity per sample; PD) for *M. myotis* was 22.34. Similar PD values were found between male (20.4) and female (23.85) and between juveniles and adults (Figs. 1A and 1B; Table 2), with neither comparison showing significant differences (Kruskal–Wallis test, $p > 0.05$). Mean PD values for each data category are displayed in Table 2, with additional bacterial abundances displayed in Fig. S2 and Table S1.

A total of 47 OTUs were present in 50% of all samples consisting of the Actinobacteria, Chlamydiae, Cyanobacteria, Firmicutes, Proteobacteria and Tenericutes phyla highlighting some inter-individual OTU diversity (Fig. S3), with Proteobacteria and Chlamydiae/Tenericutes having the highest and lowest abundance of OTUs, respectively. When increasing this core microbiome threshold, 13 OTUs, containing *Ureibacillus* (Firmicutes), *Corynebacterium* (Actinobacteria), *Enterococcus* (Firmicutes) and *Pseudomonas* (Proteobacteria; Table 3, Table S2, Fig. S4), were found present in 80% of all bats.

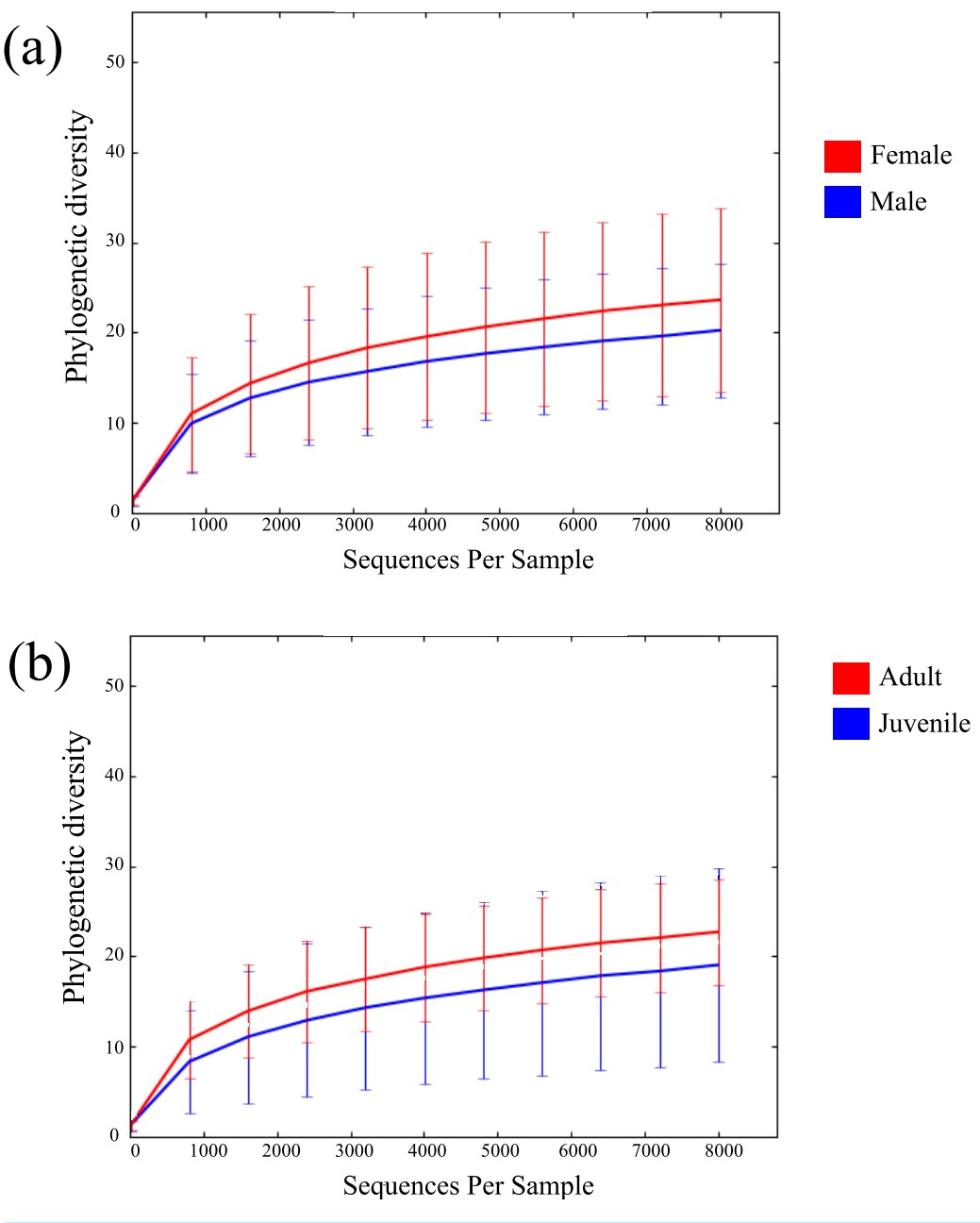

**Figure 1** **Rarefaction curves of bat microbiomes.** Read data was rarefied to 8,000 OTUs, and measured using phylogenetic diversity. (A) The alpha diversity of bat samples were clustered based on gender to investigate differences in bacterial abundance between males and female. (B) Reads were clustered based on age datasets, with age dataset 2 displayed (0 yrs considered juvenile, ≥1 yr considered adult).

## Statistical analyses

Principal coordinate analysis showed no clear observable separation of samples based on age, gender or collection site for both un-weighted and weighted Unifrac distances. Weighted Unifrac distances explained a greater percentage of variance using three principal coordinates both before rarefaction (55.27% variance, Figs. S5, S6) and after rarefaction (60.59% variance, Figs. 2A, 2B; Fig. S7). Using ANOSIM, no significant differences were found after Bonferroni correction, reflected in the low R test-statistics ($R \leq 0.14$; *Chapman & Underwood, 1999*) in all cases. A Kruskal–Wallis non-parametric ANOVA found no significant differences between male and female bats, ruling out gender as a mode of microbial diversity. OTUs were also compared between adult and juveniles (age dataset 2, 3) using a Kruskal–Wallis test revealing four OTUs from the common GI-inhabiting family *Helicobacteraceae* showing differential abundances after Bonferroni correction ($p < 0.05$) in age dataset 2, and no differentially represented OTUs observed between juvenile and adult bats for age dataset 3.

When comparing the percentage dissimilarity of OTU abundance between sites, the average dissimilarity across sites for phyla present was 52.68% and 53.79% before and after rarefaction, respectively. At this taxonomic level, the phyla that contributed most to the overall dissimilarity were Firmicutes (mean dissimilarity 15.09% before and 16.68% after rarefaction) and Proteobacteria (mean dissimilarity 14.83 before and 16.45% after rarefaction). Dissimilarity increased for each taxonomic level, with 80.42% and 85.3% dissimilarity at the family level (Table S3). This dissimilarity may reflect the absence of specific OTUs in one site relative to another, however such dissimilarity did not have a strong affect overall when comparing Unifrac distances, as site-specific clusters could not be determined using PCoA (Figs. S6, S7). Only one OTU, from the phylum Actinobacteria, showed differential abundances between locations (ID: 179312; $p = 0.0002$) suggesting that the collection site did not have a significant effect on microbial composition.

## PICRUSt and phylogenetic comparison to other mammals

Across all samples, PICRUSt inferred a total of 41 KEGG pathways present in the *M. myotis* anal flora (Table S4; Fig. 3). Of these 41 pathways, the majority of genes and their top two pathways belonged to membrane transport (14.24%; 'transporters' and 'ABC transporters'), carbohydrate transport (10.51%; 'amino sugar and nucleotide sugar metabolism' and 'glycolysis/gluconeogenesis'), amino acid metabolism (9.35%; 'amino acid related enzymes' and 'arginine and proline metabolism'), replication and repair (7.62%; 'DNA repair and recombination proteins' and 'chromosome') and energy metabolism (5.21%; 'oxidative phosphorylation' and 'carbon fixation pathways in prokaryotes').

When comparing predicted metagenomic function between juvenile and adult bats, it was found that out of 2,330 KEGG pathways analyzed, 1,713 predicted functional categories were 'enhanced' between juvenile and adult bats for age dataset 2, with a large number of 'metabolic processes' in both the enhanced (726) and divergent (246) classification across age groups (Table S5A). Enhanced categories indicate that the functional frequency, rather than the contributing microbiome community, differs across age while divergence implies

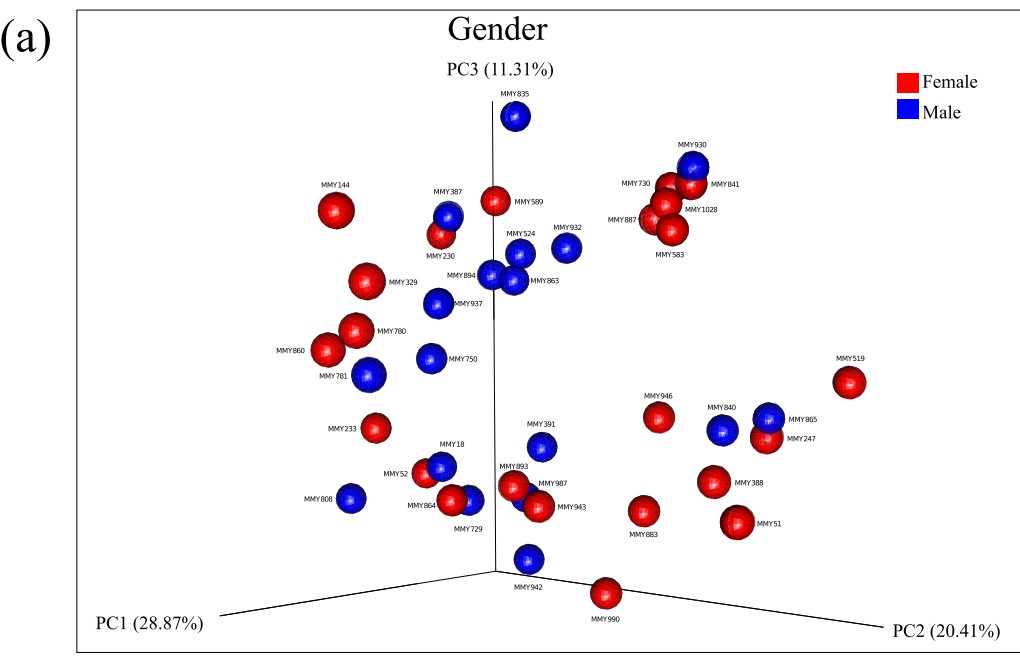

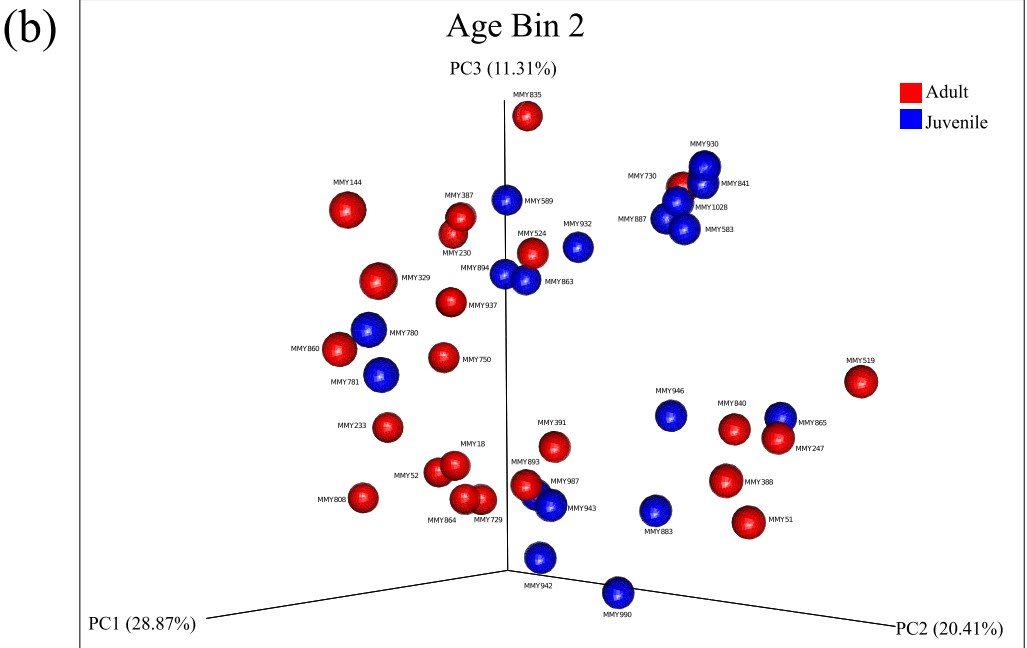

**Figure 2 Principal components analysis of beta diversity.** Similarity based on diversity between samples was explored using PCoA after rarefaction, explaining 60.59% variance. (A) Beta diversity of male and female samples using weighted Unifrac distances are displayed. (B) Beta diversity using weighted Unifrac distances between adults and juveniles in age dataset 2 are displayed.

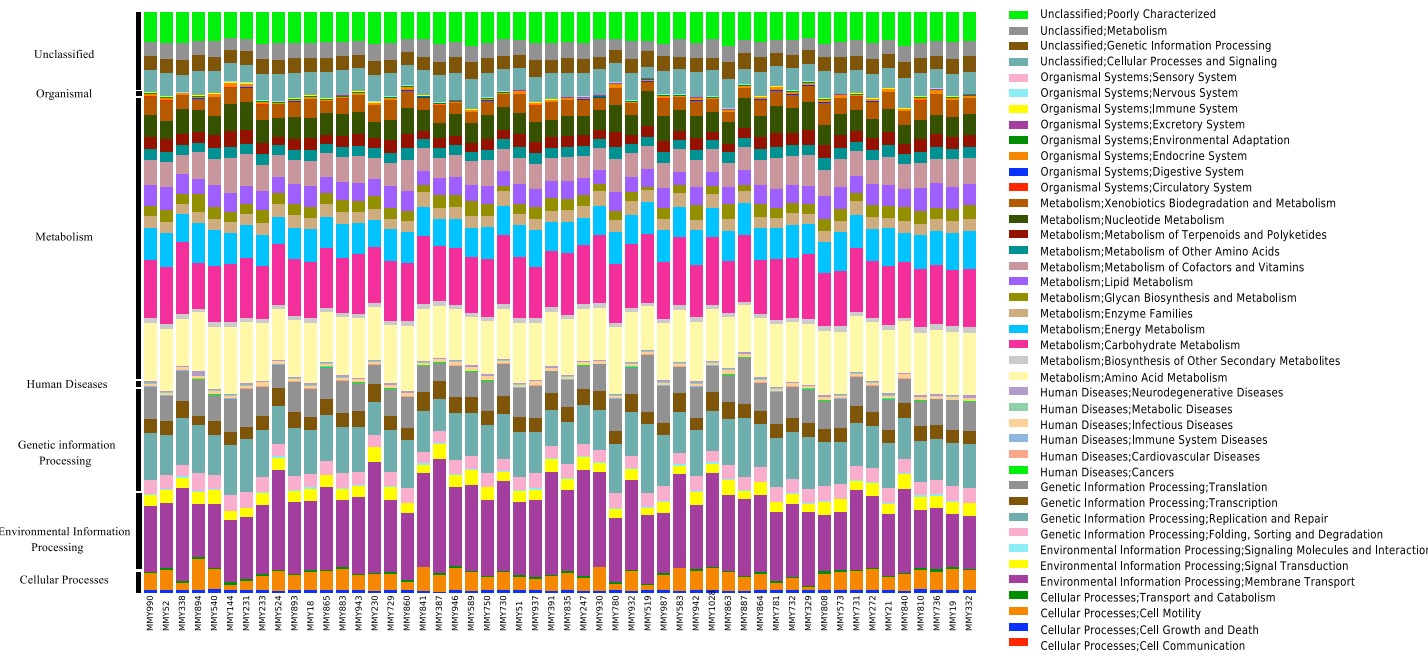

**Figure 3  KEGG pathways present in the bat anal microbiota.** PICRUSt analyses of the bat anal microbiota identified a number of different KEGG gene pathways present.

both the function and contributing community are different. Only enhanced functions were found for age dataset 3, the majority of which were involved in metabolism (Table S5B).

A comparison of 44 mammal microbiomes, including *M. myotis*, highlighted high abundances of Proteobacteria, but lower abundances of Bacteroidetes in bats compared to terrestrial mammals (Fig. 4). These data were analyzed and visualized using a PCA plot to investigate if closely related species shared similar microbial composition (76.8% variance; Table S8). Using a *k* value of 4, two clusters composed almost exclusively of bats (Table S6) were identified. Separation between bats and other mammals was apparent, with a large number of bat samples driven by the presence of Proteobacteria (Cluster 3; Table S6), despite differing diets (Fig. 4).

## DISCUSSION

When analyzing the anal microbiome of *Myotis myotis*, phylogenetic diversity was slightly higher (22.34) than the PD of the gut microbiome previously established in Vespertilionidae (19.055; *Phillips et al., 2012*). As these data were obtained through non-lethal sampling, this approach will allow re-sampling of the same individual each year. It is therefore an extremely useful tool for longitudinal microbiome studies. The mean number of observed OTUs was similar between 0 and 1 year olds (age dataset 1) and did not reflect increases observed for similar ages in humans (*Yatsunenko et al., 2012*). An increase in gut flora diversity between nursing and weaning juveniles has also been observed in pigs (*Frese et*

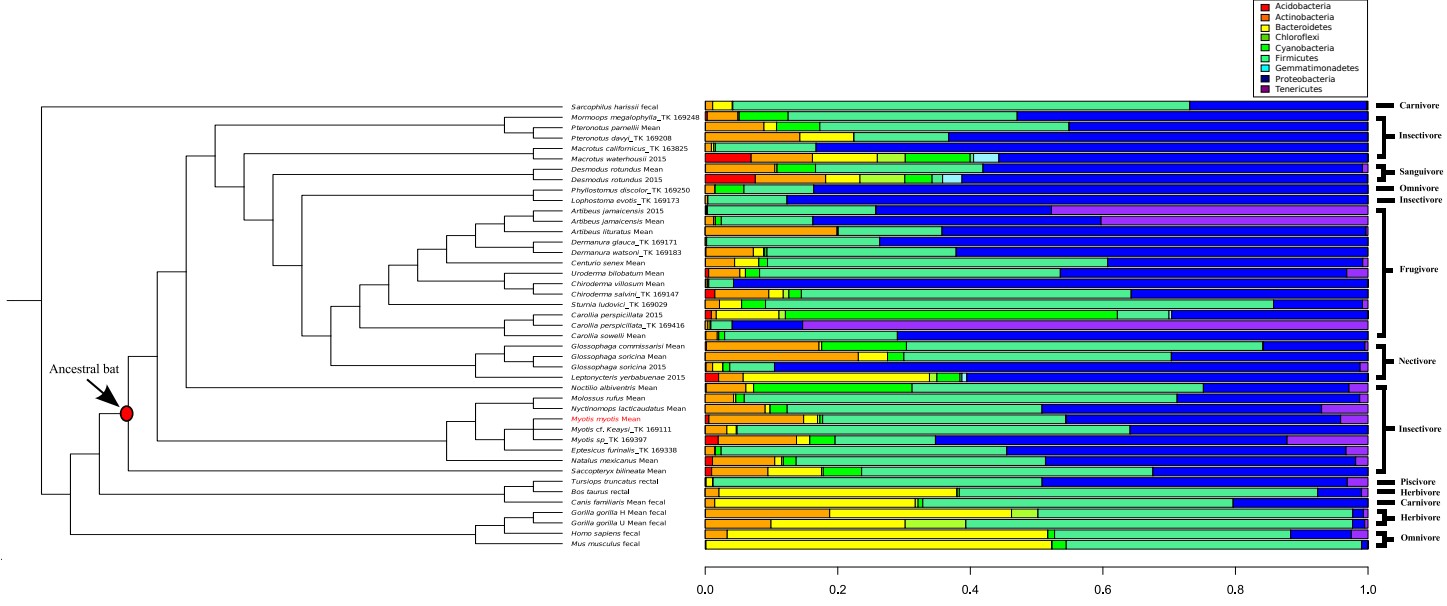

**Figure 4** **Phylogenetic comparison of microbiome bacterial abundance.** The most abundant bacterial phyla expressed in a range of diverse mammalian gut microbiome samples are compared and displayed using an established mammalian phylogeny. Bat samples from *Carrillo-Araujo et al. (2015)* are identified with '2015'. Dietary niches are also displayed.

*al., 2015*). It was not possible to determine weaning/nursing status of juvenile bats sampled in this study, however, as all juveniles were flying, there were likely partially weaned.

In their study of the aging mouse microbiome, *Langille et al. (2014)* observed a clear and statistically significant separation between young (0.48 years), middle (1.61 years) and old-aged (2.35 years) mice based on gut flora. Using human data, *Yatsunenko et al. (2012)* and *Biagi et al. (2016)* identified clustering patterns based on a variety of age ranges, spanning 0–83 and 22–109 years old, implying a general trend in aging. Using similar methods, despite some variability across individuals, we were unable to identify any clear separation between juvenile and adults in 41 *M. myotis* bats, suggesting an element of stability in microbial diversity. The oldest definitive age in our sample cohort is 4 years, representing the MLS of a mouse, a similar sized mammal. The lack of samples representing the late stages of the 37-year lifespan precludes inferences about microbiome composition in the oldest *M. myotis* bats relative to juvenile and adult bats. However, if the microbiome of *M. myotis* remains stable or static throughout their life, it is expected that the abundances of flora in 'old' individuals will be similar to the juveniles and adults described here, further implying microbiome stability as they age. This would emphasize the importance of microbiome stability in longer health-spans, resembling the fine tuned level of microbiome homeostasis regulating host aging in *C. elegans* (*Han et al., 2017*). Our intention in future studies is to involve older *M. myotis* bats to further elucidate the potential role of the microbiome in exceptional longevity and to what extent microbial variability between samples affects global microbiome patterns in bats.
The finding of dominant and variable abundances of Firmicutes and Actinobacteria is consistent with previous mammalian microbiome profiles (*Ley et al., 2008*). The genera identified in 80% of samples belong to families with many pathogenic species. As these taxa could not be identified to species or strain level, no conclusions can be drawn regarding their presence in the core *M. myotis* microbiome. Despite differences in OTUs found across sites, no significant effect of locality and thus roost specificity could be established. Bacteria from the genera *Citrobacter*, *Enterobacter*, *Escherichia*, *Klebsiella*, *Proteus* and *Streptococcus* were found present in a number of samples, and have previously been documented in the feces of *M. myotis* in Italy (*Di Bella et al., 2003*). A study of the anal microflora in *M. myotis* in Poland identified species from the *Lactobacillus*, *Enterococcus*, *Serratia*, *Corynebacterium* and *Pseudomonas* genera (*Rózalska et al., 1998*), all of which are represented in samples included here, indicating a common core of flora, independent of location. A common microbial core is similar to that observed in ruminants (*Henderson et al., 2016*), and contrasts to the social effects of microbial diversity observed in wild baboons (*Tung et al., 2015*).

The characterization and comparison of metagenomic content in the anal microbiome determined that the frequency of a number of known functional categories, of which a large proportion were involved in general metabolism, were enhanced from juveniles to adults. There was some divergence observed for age dataset 2, implying some difference in functional abundances across certain samples (1 year old). Other categories involved in energy consumption, DNA repair and oxidative phosphorylation were also present and enhanced and may imply a role of bat gut flora in enabling them to counteract the deleterious effects of metabolically costly powered flight (*Shen et al., 2010*). Studies of metabolism in insectivorous bats have determined that flight is fuelled directly by ingested prey (insects from the families Carabidae, Orthoptera, Diptera and Arachnida in *M. myotis* (*Di Bella et al., 2003*)) immediately after consumption, implying extremely rapid metabolism (*Voigt et al., 2008*; *Voigt, Sörgel & Dechmann, 2010*). This has also been observed in nectivorous and frugivorous bats (*Voigt & Speakman, 2007*; *Amitai et al., 2010*). A high abundance of bacteria producing enzymes involved in energy-related pathways might contribute to the overall managing of, and coping with, by-products of such high metabolism in *M. myotis*. If this is the case, a relatively stable microbial community, as observed between adults and juveniles, may not only be involved in extended longevity, but also play an important role in sustained flight.

By comparing microbiome content across 44 mammals, certain 'bat-specific' clustering was observed. However, without longitudinal data, investigating if the abundance of bacteria in other bats species changes with age, like humans and mice, is not possible at this time. An abundance of 'Proteobacteria' was observed in multiple different bat species, including *M. myotis*. Interestingly, Proteobacteria also show high abundances in birds (*Hird et al., 2015*), suggesting this phylum or the metabolic by-products it produces, may have a putative role in flight and its metabolic costs rather than aging. As the abundances of the bacteria phyla in the *M. myotis* microbiome appear similar to other bat compositions, it is possible that similar gene pathways are expressed across Chiropteran metagenomes, implicating a putative role of the microbiome in dealing with the energetic demands of

flight. Terrestrial, non-chiropteran species had a much larger abundance of Bacteroidetes relative to bats. The reported low abundance of the Bacteroidetes phylum in the bat gut is consistent with the same results from bat skin (*Avena et al., 2016*).

## CONCLUSION

Previous studies investigating the role of the microbiome in aging have focused on human and mice samples, and have demonstrated correlations between microbial changes and a pathogenic aging phenotype. Given the exceptional long life and health-span observed in certain species of bats given their body size, we have investigated the microbiome of the long-lived *M. myotis* using juvenile and early adulthood samples, covering the lifespan range of a similar sized mouse, acquired using a non-lethal mode of sampling. Consistent with earlier studies of bat gut flora, we find a high level of Proteobacteria and Firmicutes, with the microbiome possibly contributing to metabolism, DNA replication/repair and oxidative phosphorylation. Despite some variability across samples, distinct differential abundances in bacterial composition between adult and juvenile bats were not found, contrasting to patterns observed in humans and mice. Instead we observe an element of microbiome stability between juvenile and adult *M. myotis* bats. Given the KEGG pathways present in the anal bat flora, it is possible that metabolites produced by the bat microbiome enable them better tolerate the damaging by-products of flight and may increase metabolic efficiency, with downstream affects on aging. Future studies of older *M. myotis* age cohorts will determine whether patterns observed here continue into old age, and to what extend inter-individual variability affects global microbiome patterns in bat species. Such comparative microbiome studies will help further elucidate bacterial phyla that may contribute to healthier aging.

## ACKNOWLEDGEMENTS

We thank Nicole Foley, David Jebb, Serena Dool, Eric Petit, Frédéric Touzalin, Olivier Farcy and Arnaud Le Houédec, and the numerous volunteers and students from BV and University College Dublin for their extensive help in the field, sample collection and the owners/local authorities for allowing access to the sites. We thank Renee Potens, Nidhi Vijayan and Jorie Skutas at NSU for assistance with molecular sample processing and Cole Easson for his help with read processing.

### Funding

This study was funded by a European Research Council Research Grant ERC-2012-StG311000 and a UCD Seed funding grant awarded to Emma C. Teeling, and was supported by the Contrat Nature 'Etude de la dynamique des populations de grand murin (*Myotis myotis*) en Bretagne et Pays de Loire' awarded to Bretagne Vivante. The funders had no role in study design, data collection and analysis, decision to publish, or preparation of the manuscript.

## Grant Disclosures

The following grant information was disclosed by the authors:

European Research Council Research Grant: ERC-2012-StG311000.

UCD Seed funding.

Contrat Nature 'Etude de la dynamique des populations de grand murin (*Myotis myotis*) en Bretagne et Pays de Loire'.

## Competing Interests

The authors declare there are no competing interests. John Leech is also a research scientist at the Teagasc Food Research Centre, Fermoy, Cork, Ireland.

## Author Contributions

- Graham M. Hughes conceived and designed the experiments, performed the experiments, analyzed the data, contributed reagents/materials/analysis tools, wrote the paper, prepared figures and/or tables, reviewed drafts of the paper.
- John Leech performed the experiments, analyzed the data, contributed reagents/materials/analysis tools, wrote the paper, prepared figures and/or tables, reviewed drafts of the paper.
- Sébastien J. Puechmaille analyzed the data, contributed reagents/materials/analysis tools, wrote the paper, reviewed drafts of the paper.
- Jose V. Lopez conceived and designed the experiments, performed the experiments, contributed reagents/materials/analysis tools, wrote the paper, reviewed drafts of the paper.
- Emma C. Teeling conceived and designed the experiments, contributed reagents/materials/analysis tools, wrote the paper, reviewed drafts of the paper.

## Animal Ethics

The following information was supplied relating to ethical approvals (i.e., approving body and any reference numbers):

All captures and sample collections were carried out in accordance with the ethical guidelines and permits delivered in 'Arrêté' by the Préfet du Morbihan, Bretagne awarded to Eric Petit, Frédéric Touzalin and Sébastien Puechmaille for the time period 15 June-15 September 2013–2017. Full ethics approval and permission (AREC-13-38-Teeling) for capture and field sampling was also awarded by the University College Dublin, ethics committee to Emma Teeling. Access to all field sites was granted by local authorities in collaboration with Bretagne Vivante. Shipment of samples to NSU was permitted via US Public Health Service permit # 2014-07-015 issued by the Center for Disease Control to Jose Lopez.

## Data Availability

Hughes, GM; Leech, J; Puechmaille, SJ; Lopez, JV; Teeling, EC (2018): Is there a link between aging and microbiome diversity in exceptional mammalian longevity? figshare. https://doi.org/10.6084/m9.figshare.5263567.

## Supplemental Information

Supplemental information for this article can be found online at http://dx.doi.org/10.7717/peerj.4174#supplemental-information.

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
