# Peer review of "Is there a link between aging and microbiome diversity in exceptional mammalian longevity?"

_PeerJ, doi:10.7717/peerj.4174_

## Round 0.1 · original submission · Major Revisions

Both reviewers indicated that they found merit in your study, but also listed a number of important issues regarding experimental setup, methods used as well as data analysis and interpretation, such as use of rarefaction.

Reviewer 1 ·

Basic reporting

citations are missing

Experimental design

The issues with experimental design are closely linked to the validity of findings, so these are commented on below.

Validity of the findings

This study investigates the anal microbiome of Myotis myotis collected at different localities in France. The authors frame the analysis of microbiome composition in terms of aging, as these bats can be long lived.

The authors find no significant differences between sexes or ages, and conclude that this species has a static anal microbiome and that this is relevant to aging. They also comment on the abundance of Proteobacteria and certain functional categories that were inferred and how they think these observations are relevant to aging.

There were multiple aspects of the interpretation and data presentation of this article that are problematic. One of the most obvious flaws is that the microbiomes are said to be 'stable' over time. In the discussion it is stated that microbiomes are "unchanging" between ages. Although not explicitly presented, it appears, for example from consideration of the range in number of OTUs observed from sample to sample, that there is considerable variation in this dataset (interindividual). How does one reconcile the wide latitude in differences among individuals with the notion that there is an aspect of stability that helps protect bats from aging?

How does the fact that the dataset appears to only include bats that are not at the upper end of the life span influence the ability to design a robust study to look at microbiomes in bat aging? It seems that it would preclude the ability to address the question, and that the comparison of results in humans and mice is not valid. Given that these bats appear to roost in groups, how could microbiome dispersal effects arising from social interactions (Elife. 2015 Mar 16;4. doi: 10.7554/eLife.05224.) erase any age-related patterns? Bats were collected at different localities, but the effect of locality, was not assessed.

If only some bats are considered to be long lived (if this is the case), how is the observation that Proteobacteria is apparently a bat-wide association relevant to aging?

The discussion of functions inferred from the metagenomes appears to only include a list of general functions that are the most abundant in bacterial genomes in many datasets. I don't see any function frequencies that stand out from the norm for a digestive microbiome, although the authors conclude that the presence of metabolic pathways, among others, is somehow signal for the specific bat-bacteria energetic relationship. It doesn't seem likely that tabulating the frequency of KEGG terms summarized at some ontological level, and summarized across all bacteria in a microbiome, then doing a test for frequency differences between groups would be empowered to understand any age-related functional differences. It may be worthwile for the authors to consider methods presented in this recent work (Integr Comp Biol. 2017 Jun 28. doi: 10.1093/icb/icx011. ).

The authors rarefy their dataset. There has been a sound statistical justification for not doing so. Please see (PLoS Comput Biol. 2014 Apr 3;10(4):e1003531. doi: 10.1371/journal.pcbi.1003531.).

The authors state that there was "apparent separation of bat to other mammals in PCoA". Visual inspection of the plot does not support this statement (Sup. Fig. 5), but more importantly, such statements must be based on statistical testing, and this is not the case.

Although studying the occurrence of competitive exclusion in microbiomes is interesting, it doesn't seem that the putative identification of bacterial genera that belong to families that include pathogens warrents any mention of this ecological phenomenon. There is no functional analysis presented to support that observed lineages compete for niche space with known pathogens.

The conclusion of this manuscript is very speculative, and not supported by the data. For example, there appears to be no data connecting Proteobacteria, any of the observed functions, and biology that has effects on longevity.

·

Basic reporting

Graham Hughes and colleagues present an interesting article about the gut (anal) microbiota variation across age in bats. The manuscript is well written and well supported using many references, and the structure of the article is correct. Besides, I believe this is the first paper dealing with this topic in Chiropterans, thus it is a timely and original work. They found no effect of age, which contrasts with the general trend observed in other organisms where microbiota composition varies and becomes more diverse through development. I think this finding is relevant.

However, the employed methodologies (or at least the way to present them) have major limitations, and I think they are not appropriate to in depth in the interpretation of the results. I feel that many interpretations of the data and conclusions are too courageous considering the employed approach, namely DNA metabarcoding + Picrust, and I also believe the methodological aspects of the study, or at least the way of presenting them, can be considerably improved.

Experimental design

Regarding the methodology, I think there are two major issues that need modification or further explanation. First, I miss information about negative controls. I have personally used the Illumina Nextera kit and I know how easily cross-contamination and amplification of environmental bacteria can occur. Thus, it is necessary to mention how the effect of these putative problems was controlled, because cross-contamination could make samples more similar (resulting in no differences between groups) and diverse (resulting in larger phylogenetic diversity than previously reported) that what they actually are. I don’t say the results of this study are artifactual, but this experimental design definitely requires negative controls to avoid such problems.

Second, I am also concerned about rarefaction curves and sequencing depths. I can think about two intepretations of the sentence “Samples were rarefied…“ in the methods section, but both are incorrect from my point of view.

FIRST INTERPRETATION (most likely)- First the authors say that the data was rarefied to 8000 OTUs, which I think makes no sense. I think they mean they randomly picked 8000 sequences per sample to level sequencing depth to 8000 sequences. I wouldn’t say that’s rarefying but leveling. Besides, leveling to 8000 OTUs does not make sense, since the number of OTUs of each sample will show the actual diversity. It should be 8000 reads or sequences, depending on when the leveling step was carried out. I guess it was done after quality filtering, thus I would use 8000 sequences. In addition to terminology though, it is not clear how the authors assessed the sequencing depth, and if the employed sequencing depth of each sample was enough for correctly characterizing each amplicon product. They present average values of phylogenetic diversity in Figure 1, but I believe individual information should be provided. In this kind of metabarcoding studies interested in compositional and diversity differences I believe it is necessary to generate OTUs x sequences rarefaction curves for EACH sample, to estimate whether sequencing depth was enough to cover the entire diversity. This can be easily done using the R packages vegan, iNext or DivE. Finally, there are 11 samples with less than 8000 reads according to Table 1, and it is not clear what the authors did with these samples. I think they should be discarded.

SECOND INTERPRETATION (less likely) - If what the authors did was to use a rarefaction and extrapolation method (iNEXT r package, for instance) to estimate the diversity at 8000 sequences (not OTUs) regardless the samples had more or less sequences than 8000 this approach is extremely risky, since diversity estimates will have very uncertain when the number of sequences is very low.

Either case, these methodological issues should be addressed before publishing.

Validity of the findings

The validity of the findings is related to the comments mentioned above.

Additional comments

I also have some minor edition suggestions:

Abstract

“The anal microbiome was sequenced” > make clear you did metabarcoding and not shotgun sequencing, since as it is written now, it seems you have sequenced the whole microbiome, more so when you talk about function in the next sentence.

“Functional gene categories expressed in the M. myotis microbiome” suggest you did transcriptomics, which is not true. Thus, I would use “functional gene categories inferred from metabarcoding data” instead.

Introduction

Line 64: species > organisms

Line 67: make the first sentence more explicit. I suggest: “we have used DNA metabarcoding of the bacterial 16S rRNA gene” (which implies the use of HTS)

Lines 99-112: it should be written somewhere that the weaning/nursing status of juvenile bats was not possible to determine (as it is said in the discussion). Besides, in the discussion, it would be appropriate to add a sentence about the most probable feeding state of the sampled bats based on capture date and average weaning time of Myotis myotis bats.

Lines 120-122: split the library building step and the actual sequencing. It would be more appropriate to write something like: “all 52 samples were built into Illumina sequencing libraries using the Nextera XT (…) and sequenced in a MiSeq platform using 250PE chemistry”

Line 123: mention which scripts from Fastx toolkit were used for the quality filtering

Line 132: I know Picrust requires Greengenes, but according to my personal experience the taxonomy assignment success is larger using Silva, and ideally different databases should be combined (Silva, RDP, Greengenes, NCBI…) (just a comment for the authors).

Line 137: mention the software/script used for rarefaction/leveling/extrapolation…

I hope the authors will address all the issues raised in this report and the improved manuscript will be published in PeerJ, as I think it will be a nice contribution to the literature.

All the best and good luck
Antton Alberdi
University of Copenhagen

---

## Round 0.2 · accepted · Accept

Very interesting paper that certainly got stronger after incorporation of the reviewers' suggestions.

Reviewer 1 ·

Basic reporting

No problems.

Experimental design

No problems.

Validity of the findings

No problems.

Additional comments

I believe that you provided a very diligent and well-rounded respond to my previous comments. Good work.

·

Basic reporting

The authors have addressed all the issues I raised in the previous review round.

Experimental design

The authors have addressed all the issues I raised in the previous review round.

Validity of the findings

The authors have addressed all the issues I raised in the previous review round.

Additional comments

The authors have addressed all the issues I raised in the previous review round.